# Various Factors May Modulate the Effect of Exercise on Testosterone Levels in Men

**DOI:** 10.3390/jfmk5040081

**Published:** 2020-11-07

**Authors:** Ruba Riachy, Kevin McKinney, Demidmaa R. Tuvdendorj

**Affiliations:** Division of Endocrinology and Metabolism, Department of Internal Medicine, University of Texas Medical Branch, Galveston, TX 77555, USA; ruriachy@utmb.edu (R.R.); khmckinn@utmb.edu (K.M.)

**Keywords:** endurance, resistance, exercise intensity, obesity

## Abstract

Exercise has been proposed to increase serum testosterone concentrations. The analysis of existing literature demonstrates a large degree of variability in hormonal changes during exercise. In our manuscript, we summarized and reviewed the literature, and concluded that this variability can be explained by the effect of numerous factors, such as (a) the use of different types of exercise (e.g., endurance vs. resistance); (b) training intensity and/or duration of resting periods; (c) study populations (e.g., young vs. elderly; lean vs. obese; sedentary vs. athletes); and (d) the time point when serum testosterone was measured (e.g., during or immediately after vs. several minutes or hours after the exercise). Although exercise increases plasma testosterone concentrations, this effect depends on many factors, including the aforementioned ones. Future studies should focus on clarifying the metabolic and molecular mechanisms whereby exercise may affect serum testosterone concentrations in the short and long-terms, and furthermore, how this affects downstream mechanisms.

## 1. Introduction

Testosterone is the most potent naturally secreted steroid androgenic hormone. It is required for promotion of secondary male-sex characteristics, as well as muscle growth and neuromuscular adaptation [1]. At the muscle level, testosterone is known to exert its anabolic effect via the following two mechanisms: (a) stimulating amino acid uptake and protein synthesis; and (b) inhibiting protein degradation by counteracting cortisol signaling [1]. Age, higher body weight, poor nutritional status, stress, sleep deprivation, and alcohol consumption are known physiological factors leading to lower serum testosterone concentrations. [2,3]. Low plasma testosterone concentration is associated with fatigue, sexual dysfunction, depressed mood, difficulty concentrating, and hot flashes [4]. If left untreated, patients may develop anemia, low bone mass density (i.e., osteoporosis), higher pro-atherogenic lipoprotein-associated changes [5], and muscle wasting [4]. Thus, maintaining physiological levels of testosterone has significant health benefits.

Exercise has significant health-related benefits and it is proposed to increase plasma testosterone concentrations [6]. However, analysis of the existing literature demonstrates a large degree of inter-individual and inter-study variability in hormonal changes during exercise. Age, body weight, and exercise type, together with exercise intensity, volume, and the involved muscle type, were studied as factors modulating these hormonal changes. This review article intends to clarify the factors that contribute to the variability in serum testosterone concentrations during exercise, and the underlying mechanisms. Part 1 will focus on the acute or immediate post-exercise changes in plasma testosterone concentrations, and Part 2 will discuss the changes in basal or resting plasma testosterone concentrations after completion of exercise protocols.

An online search through the Pubmed and Medline databases was initially performed using the combination of the following keywords: “testosterone”, “exercise”, and “men”. Additional exercise description such as “type” and “intensity” were added. “Obesity” and “age” as key words were added during advanced search for population-focused data. The exclusion criteria included: publications written in languages other than English, publications involving subjects with chronic medical conditions, such as congestive heart failure and diabetes, and publications involving subjects on testosterone replacement. Table 1 contains a brief summary of the analyzed data on the effect of exercise on serum testosterone concentrations.

## 2. Part 1: Acute or Immediate Post-Exercise Testosterone Response

The changes in plasma testosterone concentrations during exercise may depend on multiple factors. Below we analyze the effect of the type of exercise (i.e., endurance or resistance), intensity, volume (i.e., the amount of muscle involved), obesity, and age on the acute or immediate post-exercise plasma testosterone changes.

### 2.1. Endurance Exercise

Endurance or aerobic exercise refers to any type of cardiovascular conditioning where breathing and heart rates increase for a sustained period of time. Although different types of endurance exercises have been performed, running and ergometer cycling with different protocols were most often used. Regarding the study populations, the majority of research done using running as a study approach was conducted in elite athletes. The study subjects in ergometer cycling studies included elite athletes, moderately active people, sedentary people, and people with obesity. Some studies have been conducted in patients with type 2 diabetes or chronic heart failure; however, these will not be discussed within the scope of this work.

Intensity refers to the load used for a given exercise. There appears to be a relative exercise intensity that must be reached in order to induce changes in serum testosterone concentrations [1]. Jezova et al. [7] compared the plasma testosterone changes during ergometer cycling conducted at three different intensities: high, moderate, and low. Significant increases in the serum testosterone concentrations were seen only in the high intensity exercise group. Kraemer et al. [8] reported that, when the number of repetitions during exercise was kept constant, the intensity determined the degree of acute post-exercise increase in serum testosterone concentrations. In the study of Kraemer et al. [9], well-trained runners underwent a treadmill running exercise with stepwise increase in intensity. The first two steps were at intermittent intensity of 60% and 75% VO2max for 10 min each (i.e., step 1 and step 2, respectively). The last two steps were at high intensity: 90% and 100% VO2max for 5 and 2 min, respectively. T-Testo increased only after 5 min of exercise at 90% VO2 max (Basal vs. 90% VO2max: 18.2 ± 3.4 vs. 24.1 ± 4.6 nmol/L, *p* < 0.01), and remained elevated at 100% VO2, returning to baseline 1 h into recovery [9]. These data support the notion that intensity is required to alter plasma testosterone concentrations. Galbo et al. [10] reported that, in young healthy men, a significant increase in T-Testo (~31%) was observed after 40 min of maximum intensity (reflected by the individual’s highest oxygen uptake) during exhaustive treadmill running. However, other factors may also affect this hormonal change. For example, in the above-mentioned study by Kraemer et al. [9] the increase in T-Testo was at 25 min after the start of the exercise, and thus it is possible that not only the intensity but also the duration of the exercise contributed to these results. Moreover, Maresh et al. [11] studied the same individuals under four different conditions: 70% and 85% VO2max intensities with pre-existing euhydration (i.e., EU70 and EU85) and hypohydration (i.e., H70 and H85). The results of their studies demonstrated that only EU85 resulted in increased T-Testo, suggesting that levels of both intensity as well as hydration are important in determining the outcomes of an exercise intervention. These results suggest that intensity among other factors can play a role in the immediate changes in serum testosterone concentrations with endurance exercise.

### 2.2. Resistance Exercise

Resistance exercise, also known as strength and weight training, involves the voluntary activation of specific skeletal muscles against some form of external resistance. This external resistance is provided either by free weights, or a variety of exercise machines [12]. Examples include heavy weightlifting, jumping, or sprinting.

Multiple studies have shown that resistance exercise can cause acute changes in serum testosterone concentrations. Circulating T-Testo has been shown to increase immediately after a bout of heavy resistance exercise and return to baseline or even decrease beyond that level within 30 min post-exercise [13]. A major determinant for this increase in plasma testosterone concentrations is the muscle mass used. Involvement of a small muscle mass, even when vigorous resistance exercise is performed, does not elevate serum testosterone concentrations above resting levels [14]. In a study of young untrained men [15], unilateral biceps curl exercise alone did not induce a significant change in post-exercise serum testosterone concentrations. However, the addition of bilateral knee extensions and leg press to the biceps curl protocol resulted in a significant elevation of the T-Testo. Shaner et al. [49] evaluated the hormonal changes with similar lower body multi-joint movement free (i.e., squats) or machine weight (i.e., leg press) exercises. Free weight exercises induced a greater increase in plasma testosterone concentrations than did the machine weight exercises. A potential explanation for this finding is that squatting requires balancing on two feet with substantial engagement of stabilizing and core musculature, such as the abdominals and back. Research on muscle activation has also shown that free weight exercise results in a greater muscle activation than machine exercise, likely by inducing a larger overall muscle mass involvement than similar machine-based weight exercises [50]. These findings are supported by studies in junior Olympic-style weightlifters [16]. Weightlifting, an example of resistance exercise involving large muscle mass, resulted in a significant elevation of T-Testo at 5 min after exercise (Basal vs. Post-exercise: 16.2 ± 6.2 vs. 21.4 ± 7.9 nmol/L). Another possible explanation for this hormonal change is the involvement of larger muscle mass, which, in addition to resistance, may be required to induce significant acute changes in plasma testosterone concentrations.

Not much has been reported about the effect of variable vs. constant exercise intensities on serum testosterone concentrations. However, Charro et al. [17] reported that, when the total volume of the load lifted is fixed, both the variable and constant exercise intensities produce similar acute changes in T-Testo in healthy young men. Similar effects were observed in healthy elderly men [51]. Another study investigated the effect of a combination of exercise intensity, muscle volume (i.e., number of sets and repetition per a set), and the duration of the resting period between the sets on the acute hormonal variations [18]. The results demonstrated that a combination of a moderate intensity, higher volume, and shorter resting periods between sets can acutely and significantly increase the post-exercise T-Testo. Interestingly, the testosterone concentrations remained elevated for 48 h after exercise cessation. Similar findings were also reported by Kreamer et al. [52] further confirming the importance of a combination of various factors to mount significant increases in the post-exercise concentrations of serum testosterone.

Lastly, Tremblay et al. [19] evaluated the effect of baseline physical activity status on hormonal changes after an exercise protocol. They studied sedentary, endurance trained, and resistance trained individuals performing endurance and resistance exercise sessions a week apart. T-Testo increased in all groups after both types of exercise sessions, but the increase was more pronounced after the resistance training. Comparing the three groups, resistance-trained individuals had a higher increase in T-Testo, especially after resistance exercise. In summary, resistance exercise appears to be a direct stimulant to testosterone production when sufficient muscle mass load is met, or when a moderate and higher exercise intensity is combined with larger muscle volume and shorter resting periods between the sets.

### 2.3. Sustainability of Post-Exercise Elevated Levels of Testosterone, and Underlying Mechanisms

Both endurance and resistance exercise studies have demonstrated an increase in plasma testosterone concentrations upon exercising; however, these levels were not sustainable beyond several minutes. In the study by Manesh et al. [11], the increase in serum testosterone concentrations was not sustained at 20 min into the recovery. Daly et al. [20] likewise showed that, despite rapid increases in T-Testo after 30 min of endurance running, the levels significantly decreased 90 min into the recovery. Several studies have sought to investigate the potential underlying mechanisms that may explain these outcomes. Cumming et al. [6] showed that both testosterone and luteinizing hormone (LH) synchronously peaked at 20 min of progressive intensity exercise on an ergometer. However, a 45 min physiological lag between the LH pulse and testosterone production was well established [53], and thus LH stimulation may not be the mechanism responsible for the increase in plasma testosterone concentrations with exercise. Aerobic exercise can provide a large physiological stress to the body, resulting in a corresponding response of the neuro-endocrine system. This can be manifested by an initial rise in plasma testosterone concentrations secondary to a catecholamine surge and testicular stimulation, followed by increases in cortisol levels, a hormone that inhibits testosterone production [54]. Others have demonstrated that an increase in serum testosterone concentrations is not secondary to increased production rate [55,56]. Cadoux et al. [54] injected radiolabeled testosterone in men who underwent vigorous aerobic exercise for 50 min. The increase in plasma testosterone concentrations during the exercise was associated with a decrease in estimated hepatic plasma flow, metabolic clearance, and plasma volume. Moreover, infusion of lactate in rats, which is normally increased during exercise, was associated with a simultaneous increase in serum testosterone concentrations, particularly by increasing the testicular cAmp production [57]. Therefore, even though aerobic exercise seems to induce the initial rise in T-Testo, this effect is not sustained due to several factors including (a) sympathetic nervous system stimulation and the subsequent inhibition of testosterone production; and (b) changes in the testosterone metabolism. In contrast, anaerobic exercise-induced stimulation of testosterone production was explained by the effect of the anaerobic glycolytic pathway on the release of gonadotropin releasing hormone (GnRH) and LH [58]. The sustainability of exercise-induced elevation of testosterone concentrations may not vary between endurance and resistance exercise; however, the underlying mechanisms may be different.

### 2.4. Obesity

So far, we have discussed works done in lean, young men; however, body weight and aging are inversely related to serum testosterone concentrations [59]. In this and the following, section, we will discuss how obesity and aging, respectively, affect the exercise-induced changes in testosterone concentrations. The underlying mechanisms of this relation include: (a) attenuated amplitude of LH pulses due to obesity-induced systemic inflammation [60]; (b) increased aromatization of circulating testosterone in the adipose tissue [61]; and (c) higher leptin production by fat cells which has been shown to disrupt testosterone production [62].

Studies investigating the effect of exercise on serum testosterone concentrations in overweight and obese individuals also show conflicting results. Rubin et al. [21] compared resistance exercise-associated serum testosterone changes in physically active lean vs. obese men. Although T-Testo was comparable between obese and lean individuals immediately after exercise, the levels were lower in the obese men (lean vs. obese: 20 vs. 8 nmol/L, *p* < 0.01) at 30 min into recovery. Despite the initially similar change in both groups, the baseline and integrated concentrations during recovery appeared to be inversely associated with the degree of adiposity. Sheikholeslami-Vatani et al. [22] investigated the acute effect of different resistance exercise orders on serum testosterone concentrations in untrained normal weight and obese men. Although in both groups T-Testo increased acutely post exercise, the increase was higher in the lean individuals, suggesting an obesity-associated blunting in hormonal changes with exercise. Another study by Velasco-Orjuela et al. [23] evaluated the acute effect of high-intensity, resistance, or combined exercise protocols on T-Testo in inactive overweight men. Surprisingly, none of the exercise protocols affected the T-Testo. The authors hypothesized that obesity-associated endothelial dysfunction and impaired vasodilatation in the testes may be, at least partially, responsible for the altered overall endocrine response in this population. Thus, whether exercise can still potentiate testosterone spikes in overweight and obese individuals is uncertain; and if present, they seem to be inferior to those seen in lean/normal weight men.

### 2.5. Age

Aging is associated with declining levels of serum testosterone concentrations in men. This is secondary to the decreasing capacity of aging Leydig cells to produce testosterone in response to LH stimulation [63]. Low plasma testosterone concentrations are associated with a number of adverse health consequences including loss of muscle mass, increased fat mass, reduced physical performance, and increased cardiovascular disease risk [64]. The effect of exercise on serum testosterone concentrations in older men is not clearly understood. Studies in older participants refer to studies in men with an average age of 60 ± 5 years. Zmuda et al. [24] examined the acute effect of moderate physical activity with increasing intensity on T-Testo in elderly (70 ± 4 years), sedentary men. Levels acutely increased during exercise and peaked at higher intensities (basal vs. post-exercise: 11.2 vs. 16 nmol/L, *p* < 0.01); however, the levels returned to baseline at 60 min into recovery. Kraemer et al. [25] examined the acute effect of heavy resistance exercise on T-Testo in young (29.8 ± 5.3 years), and older (62 ± 3.2 years) men. Both groups showed significant increases in serum testosterone concentrations immediately and 5, 10, and 15 min post-exercise. However, the magnitude of increase was higher in the younger population. Another study evaluating the effect of aging on changes in serum testosterone concentrations included three groups of healthy untrained men: young (20–26 years), middle aged (33–58 years), and older (59–72 years), who performed one session of resistance training. All three groups exhibited an increase in T-Testo post exercise, with middle aged and older men showing similar relative testosterone concentration changes to younger men. Levels returned to baseline 15 min after exercise cessation in all groups [26]. Arazi et al. [27] studied young and middle-aged men who underwent an 8-week-long progressive resistance training program. The serum testosterone concentrations were measured at four separate time points: (1) before any exercise was conducted (baseline); (2) immediately after one bout of exercise, but before the training started (i.e., pre-training immediate; I-preT); (3) immediately after completion of 8-week-long training (i.e., post-training immediate; I-postT); and (4) basal or resting post-training (PT). Compared to the baseline levels, the T-Testo concentrations were increased at I-preT in middle-aged men only. In the same group, the I-postT T-Testo were also increased and remained elevated at 30 min into the recovery. These I-postT values were also higher than the I-preT values in this group. In young men, the plasma testosterone concentrations were higher at I-preT, I-postT, and 30 min into the recovery, when compared to those of the middle-aged men. In this study, no changes were observed in the resting concentrations of serum testosterone in middle-aged men, but resting testosterone concentrations increased in young men post-training. All the analyzed studies confirmed that middle-aged and older men still mount an elevation in plasma testosterone concentrations acutely after physical activity, despite age-related hormonal declines. However, the magnitude of increase can be lower than that seen in younger men.

## 3. Part 2: Changes in Basal Resting Testosterone Levels

### 3.1. Endurance and Resistance Exercise

Many studies have addressed the effect of habitual or intervention exercise on basal (resting) serum testosterone concentrations, with no clear effect reported so far. Studies have investigated the associations between the degree of physical activity and basal plasma testosterone concentrations. The 5-year-long NHANES study included 738 participants, who were classified in three tertiles, based on metabolic equivalent of task (MET) score, and according to the compendium of physical activities. No cross-sectional association was found between a greater physical activity and changes in basal plasma testosterone concentrations [28]. Houmard et al. [29] showed that despite increasing endurance exercise’s frequency, duration, and intensity over 14 weeks (3–4 days/week, 30–45 min/day), no significant changes in the resting plasma testosterone concentrations were noted. Similarly, White et al. [30] found no change in resting testosterone concentrations with higher training mileage (i.e., 100% increase in the habitual distance run for 12 weeks) in recreational joggers. MacKelvie et al. [31] showed similar basal serum testosterone concentrations between long-distance runners and age-matched sedentary controls. In highly trained swimmers, the basal plasma testosterone concentrations did not differ between periods of intensive training and exercise tapering [32]. Interestingly, some studies have even shown that chronic endurance exercise can correlate inversely with basal serum testosterone concentrations. For instance, professional cyclists tend to have lower basal T-Testo after major competitions compared to baseline [33]. Safarinejad et al. [34] conducted a randomized trial of middle-age men undergoing intensive treadmill running. Throughout the study period, these men had low basal serum testosterone concentrations, which was associated with low follicular-stimulating hormone (FSH) and LH levels. The authors hypothesized that exercise-associated stress induced production of reactive oxygen species that can suppress the hypothalamic pituitary axis and cause hypogonadotropic hypogonadism. Interestingly, the sex hormone binding globulin levels did not decrease with declining T-Testo, reflecting that the serum testosterone changes are not related to the variation in serum binding globulin. Hackney et al. [35] reported that endurance trained men had lower T-Testo than sedentary men. In this study, LH levels were not elevated despite the lower limit values of testosterone, which may indicate HPA axis suppression with long-term endurance exercise. On the other hand, lower basal testosterone concentrations have been reported despite unaltered plasma LH and FSH levels [36]. Thus, although low basal testosterone concentration is likely due to HPA suppression and hypogonadotropic hypogonadism during chronic exercise, additional contributing factors affecting the serum testosterone concentrations in the absence of LH suppression are yet to be determined.

Although studies have proven that resistance exercise can cause significant acute changes in serum testosterone concentrations, similar changes were not observed in basal plasma testosterone levels. Nicklas et al. [37] reported no significant change in basal serum testosterone concentrations after 16 weeks of progressive resistance training program. Moreover, the previously mentioned study by Hansen et al. [15] showed unchanged resting testosterone concentrations during unilateral biceps curl exercise alone or in combination with bilateral knee extensions and leg press. Therefore, independent of exercise type, nature, or intensity, exercise does not seem to increase resting T-Testo. Based on these studies, exercise either decreases or have a neutral effect on T-Testo.

### 3.2. Obesity

The effect of exercise on basal serum testosterone concentrations in obese individuals has been evaluated in multiple studies. Although Moradi et al. [38] reported significant increases in basal serum testosterone concentrations in obese men after 12 weeks of resistance exercise (Basal vs. post-exercise: 23.9 ± 8.3 vs. 28.4 ± 5.9 nmol/L, *p* = 0.018), this correlation was not found in another study using a similar population and the same exercise type [39]. Kumagai et al. [40] investigated the effect of a 12-week aerobic exercise intervention on circulating serum testosterone concentrations in overweight/obese men. At baseline, T-Testo were significantly lower in overweight/obese men than in normal-weight men (*p* < 0.01). After the 12-week aerobic exercise intervention, serum testosterone concentrations significantly increased in the overweight/obese men (*p* < 0.01). In addition, stepwise multivariable linear regression analysis revealed that the increase in vigorous physical activity was independently associated with increased basal T-Testo (*p* = 0.011). Similarly, Rosety et al. [41] reported that a 16-week-long aerobic training program on a treadmill, with three sessions per week, increased the basal serum testosterone concentrations in obese men (baseline vs. post-test range: 15.1 vs. 16.6 nmol/L, *p* = 0.036). Khoo et al. [42] reported significant increases in serum testosterone concentrations in individuals with obesity after 24-weeks of high volume moderate-intensity exercise. Thus, most of the studies in overweight/obese men have shown a direct correlation between both aerobic and anaerobic exercise and plasma testosterone concentrations. Surprisingly, these results contradict the results of studies in lean individuals, where even those using strict protocols to stimulate acute testosterone increases failed to change basal testosterone concentrations (see above). One possible explanation for these findings is the weight/fat mass loss effect. Some studies showing direct correlations between exercise and serum testosterone concentrations also showed decreased fat mass and waist circumference in individuals with obesity [25,26]. Whether increase in basal testosterone concentrations is solely due to exercise, or is secondary to weight loss is still to be determined.

### 3.3. Age

The effect of exercise on basal serum testosterone concentrations in older men is not clearly understood. Ari et al. [43] reported that well-trained, athletic older men have significantly higher resting T-Testo than age-matched sedentary men (sedentary vs. athletic: 18.7 ± 5.9 vs. 28.8 ± 4.5 nmol/L *p* < 0.01). However, other studies were unable to distinguish differences in basal T-Testo between lifelong trained and sedentary elderly men (Hayes et al. [44] and Tissandier et al. [45]).

Some studies have been conducted to assess the changes in serum testosterone concentrations during exercise in elderly men. Hayes et al. [46] examined the impact of 6-week-long supervised exercise training on resting concentrations of serum testosterone in a cohort of lifelong sedentary men, compared to a control group of age-matched lifelong exercisers. The results revealed that only sedentary men experienced a significant exercise-induced increase in resting T-Testo. Another study by Lovell et al. [47] found no significant changes in resting plasma testosterone concentrations after 16 weeks of aerobic or resistance exercise in men aged 70–80 years. Of note, T-Testo increased immediately post sub-maximum exercise in all groups, showing a pattern similar to the post-exercise results in young adults (see above). Sellami et al. [32] conducted a randomized trial to test the effect of exercise on serum testosterone fluctuations in moderately trained young and middle-aged men (average age 20 vs. 40 years, respectively). At rest, lower T-Testo were reported in the middle-aged compared to the younger group. However, after 13-weeks of intensive anaerobic activity, the levels taken 48 h to 7 days post-exercise cessation were significantly increased in the middle-aged group, eliminating the age-associated difference between the groups. It was previously questioned whether the exercise protocols contributed to the variable results in these studies. However, even in studies involving populations with similar age, physical activity status, exercise background, and protocol duration, the change of basal plasma testosterone concentrations during exercise has not been consistent [46,47]. Based on the current literature, no conclusions can be drawn on the effect of exercise on basal serum testosterone concentrations in older men.

## 4. Discussion

This review highlights that substantial research has been done on the effect of exercise on (a) acute immediate; and (b) basal or resting post-exercise serum testosterone concentrations in men. Regardless of pre-existing conditioning, body weight, or age, sufficient evidence indicates that resistance exercise, when combined with larger muscle involvement (multi-joint movements), bigger exercise volume, sufficient intensity (moderate/high), and short resting intervals between training sets, may result in optimal acute increases in serum testosterone concentrations. However, the magnitude of this acute hormonal change is lower in older men or those with obesity.

Whether this temporary surge in post-training serum testosterone concentrations has any impact on the extent of muscle anabolism and hypertrophy is widely debated. Multiple studies found a direct link between post-exercise serum testosterone changes and muscular hypertrophic adaptation/increase in lean body mass [48]. A possible explanation is that increases in serum testosterone concentrations mediate an upregulation in acute androgen receptor expression and subsequent increases in myofibrillar protein synthesis, possibly because of enhanced ligand binding capacity or activation of the testosterone-androgen receptor signaling pathway [65]. Post-exercise peak plasma testosterone enhances androgen receptor mRNA translation and increases its half-life. Evidence suggests that acute increases in serum testosterone concentrations during exercise may likely optimize hypertrophic adaptations via enhancing the testosterone-androgen receptor [66]. On the other hand, Wilkinson et al. [67] observed significant gains in strength and hypertrophy in the absence of any measurable changes in F-Testo and insulin growth factor 1. A study by West et al. [68] showed that exposure of muscles to basal or high serum testosterone concentrations with exercise can result in similar muscle adaptations and hypertrophy. Thus, there is no solid evidence that the post-exercise acute plasma testosterone spike has a beneficial effect on muscle hypertrophy.

Data on whether exercise induces prolonged testosterone stimulation is still limited, with the majority of studies showing similar resting serum testosterone concentrations in active and inactive individuals. Some promising studies in older men have shown a direct correlation between exercise and basal plasma testosterone concentrations; however, conclusions are still preliminary until a greater depth of literature is available [44]. Similarly, studies showing positive correlations between exercise and increased basal plasma testosterone concentrations in overweight/obese individuals also showed significant associated fat loss. Whether this effect is secondary to weight loss and less aromatization, or solely secondary to exercise, is unclear.

In conclusion, the up-to-date data on the effect of exercise on serum testosterone concentrations in men have significant inter-individual and inter-study variability. This variability can be explained by (a) the use of different types of exercise (e.g., endurance vs. resistance); (b) the other factors of the training (e.g., training intensity or duration of resting periods); (c) the variety in study populations (e.g., young vs. elderly; lean vs. obese; sedentary vs. athletes); and (d) the time points when testosterone was measured (e.g., during or immediately after vs. several minutes or hours after the exercise). It is our conclusion that future studies should focus on clarifying the metabolic and molecular mechanisms whereby exercise may affect testosterone production in the short- and long-term, and furthermore how this release affects downstream mechanisms; such knowledge will be the key to understanding the exercise-testosterone-muscle hypertrophy axis.

## Figures and Tables

**Table 1 jfmk-05-00081-t001:** Summary of analyzed data on the effect of exercise on testosterone response.

Exercise Variable Factor	Population (Men)	Exercise Description	Results	Reference
Endurance	5 untrained but active Age 25 ± 1 year	60 RPM, cycle ergometer. Load increased by 25 w/min following the first 3 min at zero load, until the subject was no longer able to continue	TT significantly increased at 0, 5, 10, 15, 20, 25 min during exercise, then start declining at 25 min and no difference afterwards	[6]
Intensity, duration	7 untrained Aged 23–26 years	3 bicycle ergometer tests A, B, C of the same total work output but different intensity and duration. Tests A and B: 3 consecutive exercise bouts, lasting 6 min each, of either increasing (1.5, 2.0, 2.5 W·kg^−1^) or constant (2.0, 2.0, 2.0 W·kg^−1^) workloads, respectively Test C: 2 exercise bouts each lasting 4.5 min, with workloads of 4.0 W All the exercise bouts were separated by l-rain periods of rest	Significant increases in TT were seen only in the high intensity exercise group C	[7]
Resistance, intensity	9 healthy Age 24.66 ± 4.27 years	High resistance exercise protocols control for load (5 vs. 10 repetitions maximum (RM)), rest period length (1 vs. 3 min)	Increasing intensity load increases TT levels at 90 min post exercise	[8]
Intensity	7 well-trained runners Age: 28 ± 3 years	Strenuous intermittent exercise consisting of treadmill running at 60, 75, 90, and 100% VO2 max	TT increase IP only after 5 min of high intensity exercise at 90% VO2 max and remained elevated at 100% VO2, returning to baseline 1 h into recovery	[9]
Intensity-aerobics	8 healthy Age 20–28 years	Graded (47, 77 and 100% of maximal oxygen uptake) and prolonged (76%) exhaustive treadmill running	TT peaked at 40 min of prolonged running and then gradually declined returning to baseline 30 min post recovery	[10]
Intensity, hydration status	9 runners Age 20 years	70 or 85% VO2 max, euhydrated (U75 or U85 respectively), 70 or 85% VO2 max, hypohydrated (HY75 or HY85 respectively)	TT only significantly elevated at IP compared to pre and 20 min post exercise in U85	[11]
Muscle mass, resistance	10 resistance trained Age 18–25 years	Unilateral (dominant arm only) and bilateral upper-body RE protocol separated by 1 week, The RE protocol consisted of 3 sets of 10 repetitions of 5 different dumbbell upper-body exercises at 80% of 1-repetition maximum	TT was not affected by either protocol at 5, 10, 15 min post IP	[12]
Muscle mass	16 untrained Age 24.4 ± 3.1 years	Divided into an arm only training group (A) and a leg plus arm training group (LA). Both groups performed the same one-sided arm training for 9 weeks, twice a week	TT increased significantly in group LA but not in group A at 15, 30 and 60 min	[13]
Resistance exercise, free weight vs. machine weight	10 resistance trained men Age 25 ± 3 years	6 sets of 10 repetitions of squat or leg press at the same intensity separated by 1 week	Exercise increased TT at IP, p15 and p30 for both exercise, but significant higher increases in free weight at IP	[14]
Free weight or Smith machine squats	3 healthy males and 3 healthy females, 22 ± 1.2 years	1. set of 8RM for each of the free weight squat and Smith machine squat in a randomized order with a minimum of 3 days between sessions	EMG averaged over all muscles during the free weight squat was 43% higher when compared to the Smith machine squat	[15]
Resistance, muscle mass	28 junior elite Olympic-style weightlifters Age 17.3 ± 1.4 years	Weightlifting exercise protocol using moderate to high intensity loads and low volume, associated with the snatch lift weightlifting exercise	significant increases in TT at 5 and 15 min post exercise	[16]
Variable vs. constant intensities	10 healthy Age: 26 ± 6	Multiple-set and Pyramid with three exercises (bench press, peck deck and decline bench press) with the same total volume of load lifted	Similar testosterone responses at 30-min following each bout	[17]
Intensity, volume, resting period	10 healthy Age 21.8 ± 1.9 years	3. RE protocols included (1) H: 4 sets of 10 repetitions in the squat at 75% of 1RM (90 s rest periods); (2) S: 11 sets of three repetitions at 90% of 1RM (5 min rest periods); and (3) P: 8 sets of 6 repetitions of jump squats at 0% of 1RM (3 min rest periods)	TT increased significantly only in H group (moderate intensity, higher volume, and shorter resting periods between sets) remained elevated for 48 h after exercise cessation	[18]
Endurance vs. resistance, baseline physical activity	22 healthy 7 resistance trained 8 endurance trained 7 sedentary Age 18–55 years	3 exercise sessions completed: a resting session, endurance session (40 min run at 50–55% maximum oxygen consumption), resistance session (circuit of 7 exercises with volume matched to total caloric expenditure of the run session)	TT increased in all groups after both types of exercise sessions but the increase was more pronounced after resistance training. Comparing the 3 groups, resistance-trained individuals had a higher increase in TT especially after resistance exercise.	[19]
Endurance	22 endurance trained average age 24.6 years	Ran at 100% of their ventilatory threshold on a treadmill until volitional fatigue	Increased TT at IP for 30 min. At 90 min-24 h into recovery, TT was lower than baseline	[20]
Resistance exercise lean vs. obese	20 physically active 10 obese and 10 lean age 24.6 ± 3.7 years	RE protocol consisted of six sets of ten repetitions per leg of stepping onto an elevated platform	TT IP was similar in obese and lean, but lower in obese than lean at 30 min into recovery	[21]
Resistance exercise orders REO normal weight vs. obese	25 untrained 11 obese and 15 normal weight Age 21.73 ± 1.58 years	2 REO protocols starting with large-muscle group and progressed to small-muscle group (Protocol A), and reverse sequence (Protocol B). Each performed in 3 sets of 10 RM	TT increased IP for both protocols and in both groups, however increase in testosterone was lower in obese group	[22]
High intensity interval exercise, resistance training	51 men, overweight, physical inactivity (i.e., <150 min of moderate-intensity exercise per week for >6 months Age 23.6±3.5years	4 groups: High-intensity interval training (4 × 4 min intervals at 85–95% maximum heart rate (HRmax), interspersed with 4 min of recovery at 75–85% HRmax), resistance training (50–70% of one repetition maximum 12–15 repetitions per set with 60 s of recovery), combined high-intensity interval and resistance training, or non-exercising control	Neither exercise protocol significantly increased serum TT 1 min post exercise	[23]
Moderate physical activity/aging	7 sedentary healthy Age 66 to 76 years	4 consecutive bouts of exercise on cycle ergometer 15 min each, designed to approximate 50%, 60%, 70%, and 80% of predetermined peak heart rate reserve. Resting period 5 min between bouts	TT increased with increasing intensity declined rapidly to baseline values by 60 min after exercise	[24]
Resistance exercise older vs. young	Active (recreational sports and jogging) 8 young (30 years) 9 older (62-years)	Resistance exercise: four sets of 10 RM squats with 90 s rest between sets	Significant increase in TT in both groups at IP, 5, 15, and 30 min post-exercise, young group had a significantly higher magnitude of increase over the entire period	[25]
Resistance exercise older vs. young	healthy untrained men 8 young (20–26 years), 7 middle-aged (38–53 years), and 9 older (59–72 years)	Acute exercise protocol (3 sets, 10 repetitions, 80% of 1RM, 6 exercises)	Each group exhibited an increase (*p* < 0.05) in TT and FT immediately post exercise. middle-aged and older men showed similar relative T responses to those of younger men to a single bout of high-intensity resistance exercise	[26]
Resistance exercise young vs. middle aged	8 healthy middle aged (49 ± 2 years) and 10 young aged 21.2 ± 2.2 years	Acute moderate intensity resistance exercise testing (AMIRE) consisted of 4 sets of 12 repetitions bench press with 120 s of rest between sets performed before and after 8-week progressive resistance training program three times per week	TT increased at IP and remained elevated at 30 min post exercise for MI, with a higher level at IP compared with pre-training. Pre-training and post-training, in AMIRE, testosterone concentration in young men was higher at IP and 30P than middle aged men	[27]
Physical activity tertiles	738 participants (mean age 42.4 years, range 20–≥85 years)	Classified in 3 tertiles based on metabolic equivalent of task (MET) score according to the compendium of physical activities MET- minutes per week of total physical activity were calculated, which included the combined total METs of daily transportation, domestic and leisure-time activities	No association between PA tertiles and the odds of low or low normal testosterone	[28]
Endurance	13 Age 47.2 ± 1.5 years	Examined before and after 14 weeks of endurance-oriented physical training (3–4 days/week, 30–45 min/day)	There were no changes in basal plasma T at the end of the study period	[29]
Increasing endurance	13 fitness joggers Age 24.5 ± 2.5 years	Training intervention consisted of a 100% increase in the habitual distance run 12 miles/week for 2 consecutive weeks, while maintaining the customary training intensity	Serum basal TT concentrations did not change significantly from baseline after intervention	[30]
Volume	7 high volume runners (64–80 km/week) 5 very high volume runners (85–112 km/week) 12 nonathletic controls Age 40–55 years	Comparing basal testosterone levels between the 3 groups	Running volumes greater than 64 km a week, training was inversely related to testosterone levels, but levels remained within the normal range.	[31]
Volume	8 trained swimmers Age 21.1 ± 3.4 years	12 weeks of intensive training followed by 4 weeks of exercise tapering. Before taper, mean peak weekly amount of training reached 53,000 ± 20,000 m·week, performed in 10 sessions/week. During taper, training was reduced to 13,000 ± 8000	No change in basal TT levels during training and tapering periods	[32]
Endurance	18 professional cyclists Age 28 ± 3 years	Team 1 performed12 ± 2.7 competition days in the previous month vs. team 2 performed 6.2 ± 0.4 days. Both teams started a race of 3–4 h/day of easy to moderate bicycling	Cyclists who spent more days recently in competition had lower basal TT. During 3 week cycling competition, weekly basal TT levels tend to decrease gradually	[33]
Endurance	362 habitual aerobic exercisers Age 20–40 years	Moderate intensity exercise group (60% VO2max) and high intensity exercise group (80% VO2max) exercised for 60 weeks in 5 sessions/week, each session 120 min, followed by 36 weeks of low intensity exercise recovery period	TT and FT began to decrease starting 12 weeks in both moderate and high intensity exercise groups. Most significant decrease was at 12 weeks in high intensity exercise group	[34]
Endurance	53 endurance trained 35 sedentary Age 27.6 ± 6 years	No intervention	Basal TT and FT of the ET men were significantly (*p* < 0.01) lower than that of the SED men. The levels in the ET men where in the normal clinical range, but represented only 55% to 85% of those seen in the SED men.	[35]
Endurance	8 distance runners Age 72.5 ± 5.3 years 11 sedentary Age 69.1 ± 2.6 years	On day 1, graded cycle ergometer until exhaustion. On Day 2, cycle ergometer at vo2 65% for 30 min then infusion of corticotropin- releasing hormone, luteinizing hormone, thyrotropin-releasing hormone	Pre-exercise FT was lower in runners than sedentary group. After 30 min, FT was increased in both groups but significantly lower in the runners group	[36]
Resistance	13 Age (60 ± 4 years)	16 week progressive resistive training program including upper and lower extremities	No effect on baseline concentrations of the TT	[37]
Resistance	21 obese young in 2 groups: 10 resistance training Age 26.5 ± 2.8 years 11 Control Age 27.4 ± 2.9 years	12 weeks weight training (3 sessions per week, 10 exercises, 3 sets of 8–12 repetitions in each exercise, intensity 60–80% of one repetition maximum, rest between sets 1 min and between exercises 2 min, duration of main training 20–40 min per each session)	TT concentrations were increased after resistance training, while there were no significant changes in serum levels of these hormones in control group	[38]
Resistance	36 Age 22 years	Randomized into an resistance training (12 weeks of training, 3/week) or control group (12 weeks no training), each session lasting one hour. The training overload had 3 phases. Phase 1 (weeks 1–2), 2 sets of 12–15 repetitions for each exercise at 100% RM. In phase 2 (weeks 3–7) 3 sets of 8–12 repetitions at 100% RM, phase 3 (weeks 8–12): 6–8 repetitions at 100% RM	TT did not change in either group. These changes were noted without weight loss, and in concert with increases in lean body mass and decreasing fat mass	[39]
Aerobic exercise	16 normal-weight men and 28 overweight/obese men Age 50 ± 1.2 years	12-week aerobic exercise with physical activity levels labeled as (light, moderate, or vigorous) in all participants	TT significantly increased in overweight/obese men. In addition, stepwise multivariable linear regression analysis revealed the increase in vigorous physical activity was independently associated with increased serum total testosterone levels	[40]
Aerobic	90 obese men randomly allocated to the intervention (n = 45) or control group (n = 45) Age 25–40 years	Intervention group performed 16-week aerobic training on a treadmill, 3 sessions/week, consisting of a warm-up (10–15 min), 35–50 min treadmill exercise (increasing five minutes per four weeks) at a work intensity of 50–65% of peak heart rate and cooling-down (5–10 min)	Exercise increased basal serum testosterone levels with decrease in abdominal obesity	[41]
Low volume (LV) and high volume (HV) of moderate-intensity exercise	90 obese sedentary Asian men Age range 30–60 years	Prescribed a diet to reduce daily intake by ≈ 400 kcal and randomized to perform moderate-intensity exercise of LV (<150 min/week) or HV (200–300 min/week) (n = 45 each) for 24 weeks	The HV group had significantly greater increases in testosterone and reductions in weight, WC, and fat mass than the LV group	[42]
Chronic regular exercise	10 master athletes exercising regularly Age 68 ± 6 years 11 control sedentary Age 65 ± 5 years	No intervention	Higher basal TT levels in athletic men	[43]
Lifelong training history	20 lifelong exercisers Age 60.4 ± 4.7 years 28 sedentary Age 62.5 ± 5.3 years	No intervention	Lack of differences in basal TT between the 2 groups	[44]
Long term physical activity/age	8 trained and 9 sedentary men Age 60–65 years	No intervention	No statistical difference was observed between both groups for TT values	[45]
Conditioning exercise/age	28 lifelong sedentary Age 62.5 ± 5.3 years 20 lifelong exercisers Age 60.4 ± 4.7 years	6 weeks of conditioning training which includes walking, jogging or cycling. Lifelong exercisers maintained their regular exercise. Sedentary started first 2 weeks as moderate exercise to achieve heart rate reserve of 55% then vigorous exercise for 2 weeks to achieve hear rate reserve of 60–65% every 5 min	Lack of significant change in any parameter amongst lifelong exercisers, whilst sedentary men experienced a significant exercise-induced improvement in basal total testosterone levels	[46]
Aerobic exercise/ag	32 Age 70–80 years	Testosterone response after Bout of sub-maximum aerobic exercise before, after 16 weeks of resistance or aerobic training and again after 4 weeks of detraining	No change in resting concentrations of TT at any time point. Testosterone increased immediately post sub-maximum exercise before training, after 16 weeks training and after 4 weeks detraining with the increase in Test higher after 16 weeks of resistance training compared to before training and after 4 weeks detraining	[47]
Sprint + resistance/age	40 moderately trained assigned to a young trained (YT), young control (YC), middle-aged trained (MAT), and middle-aged control (MAC) group. Young age: 21 years Middle age: 41 years	Combined sprint and resistance training: one sprint running, one sprint cycling, and one resistance training session per week (13 sessions of each training unit). Sessions lasted no longer than 70 min Sprint running sessions: 3–5 sets of 3–5 bouts at maximum velocity Sprint cycling: 3–5 repetitions of 10–30 s Resistance training: 5–6 exercises with increasing load from 40 to 65% of 1 RM	Before exercise, lower TT was observed in middle-aged groups compared to younger ones. After exercise, basal TT increased significantly in MAT and the age-difference was absent between trained groups	[48]

TT: total testosterone IP: immediately post exercise RM: repetition maximum. NOTES: The analysis of published data demonstrated that either plasma total or free testosterone concentrations (T-Testo or F-Testo, respectively) or both are being presented. The majority of the work presents the data on either serum or plasma testosterone, while some report on the salivatory hormone. Unless specified, we will present the data mainly on T-Testo, which was the most frequently reported. Lastly, within the scope of this work, we analyzed only studies conducted in healthy men. As many conditions can alter the plasma concentration of sex hormone binding globulins, interpretation of T-Testo and F-Testo should take these conditions into consideration.

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
