# Peer review of "Various Factors May Modulate the Effect of Exercise on Testosterone Levels in Men"

_jfmk, 2020, doi:10.3390/jfmk5040081_

Round 1
Reviewer 1 Report
The authors claim that they have corrected the manuscript according to my suggestions but I am still seeing the same mistakes in the mansucript. I am afraid they have attached the old version of the manuscript and NOT the reviesed one.
Author Response
Below are the changes as suggested :
Modified all statements including Testosterone release and Testosterone response
Testosterone levels replaced by serum/plasma Testosterone concentrations
All units converted to the SI unit nmol/l
Although we totally agree with the comment regarding old studies, unfortunately the majority of these are very essential for the paper. Newer studies are not sufficient for the review. For example, Kreamer WJ has a lot of publications in this field and mostly are old ones. We had to include old studies to complete the picture.
Since hypogonadotropic hypogonadism can occur in both genders, we removed gender specification as suggested.
Reviewer 2 Report
the authors addressed all the issues
Author Response
Thank you for your comment
Reviewer 3 Report
All my comments have been addressed.
Author Response
Thank you for your comment
Round 2
Reviewer 1 Report
I still do not agree with the statement in line 268 on page 6 that "...the mechanism behind the association of low basal testosterone concentrations with chronic exercise remains unclear..."
Strenuous physical exercise, which leads to a significant decrease in fat mass will always shut down release of LH and therefore lead to hypogonadotropic hypogonadism not only in men but also in women.
I would encourage the authors to write 2-3 sentences on that matter.
Author Response
In agreement with other studies showing the low serum Tesotserone can be seen with unchanged gonadotropins, I modified the statement as below:
Thus, although low basal testosterone concentration is likely due to HPA suppression and hypogonadotropic hypogonadism during chronic exercise; additional contributing factors affecting the serum testosterone concentrations in the absence of LH suppression are yet to be determined

This manuscript is a resubmission of an earlier submission. The following is a list of the peer review reports and author responses from that submission.
Round 1
Reviewer 1 Report
Ruba Riachy and colleagues wrote a review that resumes the factors contributing to the variability of testosterone release after physical exercise in healthy men, and the underlying mechanisms. The review is very well written and easy to follow, even for lay readers.
Here I report some minor issues that should be addressed before publication:
-Page 1 line 33: testosterone levels have been recently shown to be associated to lipoprotein function (JCEM 2019, 10.1210/JC.2018-02027) rather than total cholesterol levels (on which there are more conflicting data).
- Page 2, lines 46-47: The choice of considering studies that use T-testosterone is particularly well suited, since the common assays to measure free testosterone are unreliable, and only occasionally authors correctly calculate F-Testosterone using the Vermeulen formula. However, this could mislead regarding the effect produced by conditions that alter SHBG levels such as obesity and diabetes. This should be stated in the manuscript.
- Page 2, line 49: The title “Part 1: Acute or immediate post-exercise testosterone response” should be highlited
-Page 6, lines 244-245: low T-Testo with normal gonadotropins is a common finding of adult onset hypogonadotropic hypogonadism. However, in this subset of subjects it should be assessed if the T-Testo levels are due to a decreased SHBG rather than a true hypogonadotropic hypogonadism, especially if signs and symptoms are absent (as expected in athletes)
Author Response
Point 1 :
Low testosterone levels are associated with fatigue, sexual dysfunction, depressed mood, difficulty concentrating, and hot flushes [4]. If left untreated, patients may develop anemia, low bone mass density (i.e., osteoporosis), higher pro-atherogenic lipoprotein-associated changes [ JCEM article suggested], and muscle wasting [4]. Thus, maintaining physiological levels of testosterone has significant health benefits.
Point 2 : A statement regarding SHBG was added as below
The majority of the works present the data on either serum or plasma testosterone, while some report on the salivatory hormone. Unless specified, we will present the data mainly on serum or plasma T-Testo. Lastly, within the scope of this work, we analyzed only studies conducted in healthy men. As many conditions can alter the plasma concentration of sex hormone binding globulins, interpretation of T-Testo and F-Testo levels should take these conditions into account
Point 3
Part 1 was highlighted
Point 4
The study in that paragraph included measurement of SHBG. A statement was added as below
]. Safarinejad et al. [49] conducted a randomized trial of middle age men undergoing intensive treadmill running. Throughout the study period, these men had low basal serum testosterone levels which was associated with low follicular-stimulating hormone (FSH) and LH levels. The authors hypothesized that exercise-associated stress induced production of reactive oxygen species that can suppress the hypothalamic pituitary axis and cause hypogonadotropic hypogonadism. Interestingly, the sex hormone binding globulin levels did not decrease with declining T-Testo levels, reflecting that the serum testosterone changes are not related to the variation in binding globulin levels

Reviewer 2 Report
The present manuscript reviews the factors influencing or modulating changes induced by exercise on testosterone levels. In my opinion the rationale for the review has not been clearly stated. After carefully reading the manuscript, it is not clear for this reviewer the needing for the review. Most of the information included is confused as, obviously, much more factors than the ones considered by the authors influence testosterone levels. Furthermore, it seems that for some factors, despite the authors indicate that are multiple studies related, actually there are only a few studies published, or the authors have included only a few studies. Therefore, instead of there are a lot of studies in the field, or there are multiple studies, maybe a conclusion is that there is a lack of studies regarding some of the questions considered in the present review.
Comments
Inclusion and exclusion criteria for the studies considered should be clearly stated. Furthermore, a clear bibliographic research strategy should be presented in the methods section of the manuscript: key words used in the research, number of articles found, articles discarded and reasons, etc. In spite of being a review article it also involves a methodology. Did the authors follow any systematic reviews guidelines (PRISMA or any other)?. Overall, this is an essential issue to be addressed.
Lines 46-48. Which is the reason for this? Why only total testosterone?
Line 49. I suggest highlighting this title or section. Otherwise it is even confusing.
Line 77. What is maximum intensity? It is difficult to accept that a maximum exercise intensity could be performed for 40 minutes. Everywhere details should be given in an objective way.
Lines 85-86. In my opinion much more evidence is needed to conclude that "intensity plays a significant role in the immediate testosterone release response to endurance exercise". Actually, in some of the studies reported it seems that hydration and duration play an essential role, being even more important than intensity. Therefore, in this section not only intensity has been considered.
Line 92 (and in other places). What does "direct relation" between resistance exercise and acute testosterones responses mean? Direct relation with any measurable characteristic of the exercise? Otherwise this sentence should be reworded.
Lines 222-223. It should be clarified whether the aim was to determine the effects of habitual exercise or the effect of exercise interventions, or both.
Line 224. Direct correlation between which objective measurements? Prevent using these expressions.
Lines 226-228. Include more details about participants in these studies.
Lines252-253. But the authors have reported at least three studies where decreases are shown.
Lines 255-256. Actually it has been evaluated the effect of physical activity interventions. Therefore, a lot of changes could be expected. I think it could not be considered as a proper model for accomplishing the aim of the authors. At least, the nature of the studies should have been indicated in the first sentence.
Line 284. It is not the first time that "multiple studies" is used. But then, only three studies are reported.
Table 1. Has the information in bold any special meaning? (references 16 and 20)
References. References should be checked for uniformity and adequate formatting. A lot of information (author and journal names) are underlined, as links (some of them in blue) to Pubmed.
Author Response
Point 1 : the bold part of this point is added to the manuscript
The authors did not follow a systemetic review method. A brief description about each study is outlined in table 1
Online search in Pubmed and medline databases was initially performed using the combination of the following keywords/mesh terms: “Testosterone”, “exercise”, and “men”. Additional exercise description including “type” and “intensity” were added. “Obesity” and “age” as key words were added later during advanced search for population-focused analysis. There are following exclusion criteria: publications written in languages other than English, publications involving subjects with chronic medical conditions such as congestive heart failure and diabetes, or publications involving subjects on Testosterone replacement.
Point 2
The majority of the published data in the literature involves total Testosterone, likely because of the assay preference. a statement about the importance of its interpretation given changes with SHBG was mentioned as following
Unless specified, we will present the data mainly on serum or plasma T-Testo, which was the most frequently reported. Lastly, within the scope of this work, we analyzed only studies conducted in healthy men. As many conditions can alter the plasma concentration of sex hormone binding globulins, interpretation of T-Testo and F-Testo levels should take these conditions into account
Point 3
statement highlighted
Point 4 Regarding maximal intensity
Galbo et al. [9] reported that, in young healthy men, a significant increase in levels of T-Testo (~31%) was observed after 40 min of maximum intensity (reflected by individual’s highest oxygen uptake) during exhaustive treadmill running
Point 5
Statement reformulated as below
These results suggest that intensity, among other factors, can play a role in the immediate testosterone release response to endurance exercise
Point 6
Statement reformulated as below
Multiple studies have shown that resistance exercise can cause acute changes in testosterone levels.
Point 7 and 8
Including both habitual and intervention exercise, statement formulated as below
Many studies have addressed the effect of habitual or intervention exercise on basal resting testosterone levels, with no clear effect reported so far.
Point 9 Details about participants were added as below
The 5-year-long NHANES study included 738 participants who were classified in 3 tertiles based on metabolic equivalent of task (MET) score according to the compendium of physical activities. It found no cross-sectional association between a greater physical activity and basal testosterone levels [43]. Houmard et al. [44] showed that despite increasing endurance exercise’s frequency, duration and intensity over 14 weeks (3-4 days/week, 30-45 min/day), no significant changes in the resting plasma testosterone level were noted
Point 10 and 11
Therefore, independent of exercise type, nature, or intensity, exercise does not seem to increase resting T-Testo levels. Based on the aforementioned studies, exercise either decreases or have a neutral effect on T-Testo levels
Point 12 replaced" multiple" by "some"
Some studies have been conducted to assess the response of testosterone to exercise in elderly men.
Point 13
Removed bold font from table 1
point 14
References adjusted as attached

Reviewer 3 Report
This review paper is on the effects of exercise on testosterone “release” in men. The authors reviewed the literature on this subject and concluded that this effect depends on many factors including the type of exercise and its intensity or duration, study populations and the time point when testosterone was measured. The authors have put a lot of work in this review paper however the topic of this paper is a little bit misleading.
First of all, the authors use an expression “testosterone release” throughout the manuscript. In my opinion the Leydig cells do not have a capacity of storing testosterone like beta-pancreatic cells insulin and therefore we cannot name it a RELEASE. Also, I would suggest not to use the expression of “testosterone response”. In my opinion all the observed elevations of testosterone concentrations post-exercise are due to the changes in hepatic plasma flow, metabolic clearance and plasma volume, which the authors also note on page 4 line 150.
Strenuous physical exercise, which leads to a significant decrease in fat mass will always shut down release of LH and therefore lead to hypogonadotropic hypogonadism not only in men but also in women. Therefore, the statement which appears on page 6 line 245 is NOT true.
Minor comments:
Several very old papers are quoted in this review paper: some are from the late 70’ and 80’, which means they are more than 35-40 yrs old. When writing a review paper I would suggest to concentrate on publications, over the last 10-15 yrs.
Please use the expression “serum/plasma testosterone concentrations” and NOT “testosterone levels”
When giving values of T-Testo from other studies please be consistent and use the same units (preferably SI -> nmol/L). This will allow the comparison the studies from different study populations.
Author Response
Point 1
Testosterone "release" and "repsonse" were modified in the text
Point 2
We reported "in men" since this paper does not report data including women subjects. The line also goes with your point that is it hypogonadotropic hypogonadism but doesn't dwell on the physiology behind it ( fat mass loss). It was reported as analyzed in the study the authors conduct. If you beleive the statement is misleading we can remove it
Point 3
We agree with your suggestion regarding old studies. However, some have very important data that contributed to the paper and we had to use them to complete the picture. We limited number of old studies as possible
Point 4
Modified to serum/plasma testosterone concentrations
Point 5
Reported all levels with same SI unit
